# Engineering of a mammalian VMAT2 for cryo-EM analysis results in non-canonical protein folding

Ying Lyu[1,3], Chunting Fu[1,3], Haiyun Ma[2,3], Zhaoming Su[2] ✉, Ziyi Sun [1] ✉ & Xiaoming Zhou [1] ✉

Vesicular monoamine transporter 2 (VMAT2) belongs to the major facilitator superfamily (MFS), and mediates cytoplasmic monoamine packaging into presynaptic vesicles. Here, we present two cryo-EM structures of VMAT2, with a frog VMAT2 adopting a canonical MFS fold and an engineered sheep VMAT2 adopting a non-canonical fold. Both VMAT2 proteins mediate uptake of a selective fluorescent VMAT2 substrate into cells. Molecular docking, substrate binding and transport analysis reveal potential substrate binding mechanism in VMAT2. Meanwhile, caution is advised when interpreting engineered membrane protein structures.

Vesicular monoamine transporter 2 (VMAT2) is responsible for $H^+$-dependent uptake of monoamine neurotransmitters, e.g. dopamine (DA) and serotonin (SER), from the cytoplasm to presynaptic vesicles in monoaminergic neurons[1,2]. Dysfunction of monoaminergic pathways is associated with various neurological and psychiatric disorders, including Parkinson's disease, depression and schizophrenia[3]. VMAT2 inhibitors, e.g. reserpine (RSP) and tetrabenazine, have been used to treat hypertension and Huntington's disease-associated chorea[4,5]. Furthermore, VMAT2 is also the target of amphetamines that eventually lead to vesicle deacidification and dopamine release, which underlies amphetamine addiction[6].

VMAT2 (a.k.a. SLC18A2) belongs to the major facilitator superfamily (MFS)[1,2]. MFS transporters usually contain 12 transmembrane (TM) helices, organized into N-domain (TM1-6) and C-domain (TM7-12)[7,8]. The two domains are structurally related by a two-fold pseudosymmetry perpendicular to the membrane plane. Meanwhile, each 6-TM domain is made of two 3-TM subdomains that are also related by a two-fold pseudosymmetry running parallel to the membrane plane. Substrates bind to MFS transporters between N-domain and C-domain, and are transported by a rocker-switch alternating access mechanism[7,8]. Despite accumulated knowledge for MFS transporters, structural mechanisms of VMAT2 remain to be elucidated.

In this work, we report two cryo-EM structures of VMAT2, with VMAT2 from *Xenopus laevis* adopting a canonical MFS fold and an engineered VMAT2 from *Ovis aries* adopting a non-canonical fold. Molecular docking, substrate binding and transport analysis reveal potential substrate binding mechanism in VMAT2. Furthermore, our work suggests that caution is needed when interpreting engineered membrane protein structures.

## Results

### Engineering of VMAT2 for cryo-EM analysis

To determine high-resolution structures of VMAT2, we screened expression and stability of VMAT2 from twelve species. Among them, human VMAT2 (*Homo sapiens*, HsVMAT2), sheep VMAT2 (*Ovis aries*, OaVMAT2), and a frog VMAT2 (*Xenopus laevis*, XlVMAT2) displayed good gel-filtration profiles (Supplementary Fig. 1a), and were subjected to cryo-electron microscopy (cryo-EM) single-particle analysis. However, the compact size (~56 kDa) and lack of rigid soluble domains of these VMAT2s hampered our initial efforts to resolve their structures, due to difficulties in particle alignment[9]. Nowadays, replacing flexible inter-helix loops with a well-folded soluble protein (e.g. BRIL or AmpC, see Methods for details) has become a common strategy to increase valid size and rigid soluble portion of membrane proteins for cryo-EM studies[10,11] (Supplementary Fig. 1b). Therefore, we engineered HsVMAT2,

[1]Department of Integrated Traditional Chinese and Western Medicine, State Key Laboratory of Biotherapy, West China Hospital, Sichuan University, Chengdu, Sichuan 610041, China. [2]State Key Laboratory of Biotherapy, Department of Geriatrics and National Clinical Research Center for Geriatrics, West China Hospital, Sichuan University, Chengdu, Sichuan 610041, China. [3]These authors contributed equally: Ying Lyu, Chunting Fu, Haiyun Ma. ✉e-mail: zsu@wchscu.cn; ziyi.sun@scu.edu.cn; x.zhou@scu.edu.cn

OaVMAT2 and XlVMAT2 by replacing various loops of VMAT2 with either BRIL or AmpC at different positions or with varying linker lengths (Supplementary Table 1). In the end, OaVMAT2 with the TM8/9 loop replaced by BRIL (OaVMAT2$_{TM8/9-BRIL}$) or AmpC (OaVMAT2$_{TM8/9-AmpC}$) yielded stable proteins with reasonable gel-filtration profiles (Supplementary Fig. 1c and 1d). Subsequent cryo-EM single-particle analysis revealed OaVMAT2$_{TM8/9-BRIL}$ structure at 3.2 Å resolution (Supplementary Fig. 2 and Table 1).

## Cryo-EM structure of OaVMAT2$_{TM8/9-BRIL}$

The TM region of the OaVMAT2$_{TM8/9-BRIL}$ structure was well resolved (Fig. 1a and Supplementary Fig. 3), allowing further analysis on its structural fold and substrate binding. Interestingly, the OaVMAT2$_{TM8/9-BRIL}$ structure is organized as a homodimer (Fig. 1a), which was also confirmed by chemical crosslinking using glutaraldehyde (Supplementary Fig. 1e).

Though many MFS structures are determined in a monomeric form, higher oligomeric forms of MFS transporters have been reported in membranes and in cryo-EM studies[7] (e.g. human monocarboxylate transporter 2, hMCT2[12]). The two protomers of OaVMAT2$_{TM8/9-BRIL}$ are nearly identical with an all-atom root mean square deviation (RMSD) of ~0.1 Å (Supplementary Fig. 4a). As seen in most MFS transporters, each OaVMAT2$_{TM8/9-BRIL}$ protomer contains 12 TM helices grouped into N-domain (TM1-6) and C-domain (TM7-12) (Fig. 1a). OaVMAT2$_{TM8/9-BRIL}$ N-domain adopts a canonical MFS N-domain fold (Supplementary Fig. 4b), with TM1-3 and TM4-6 related by a two-fold pseudosymmetry (Supplementary Fig. 4c). Surprisingly, OaVMAT2$_{TM8/9-BRIL}$ C-domain is folded in a unique way that is distinct from canonical MFS C-domains, which usually have TM11 and TM12 arranged near TM7 (Supplementary Fig. 4d). However, in OaVMAT2$_{TM8/9-BRIL}$ C-domain, TM11 and TM12 are positioned away from TM7, but close to TM8 and TM10

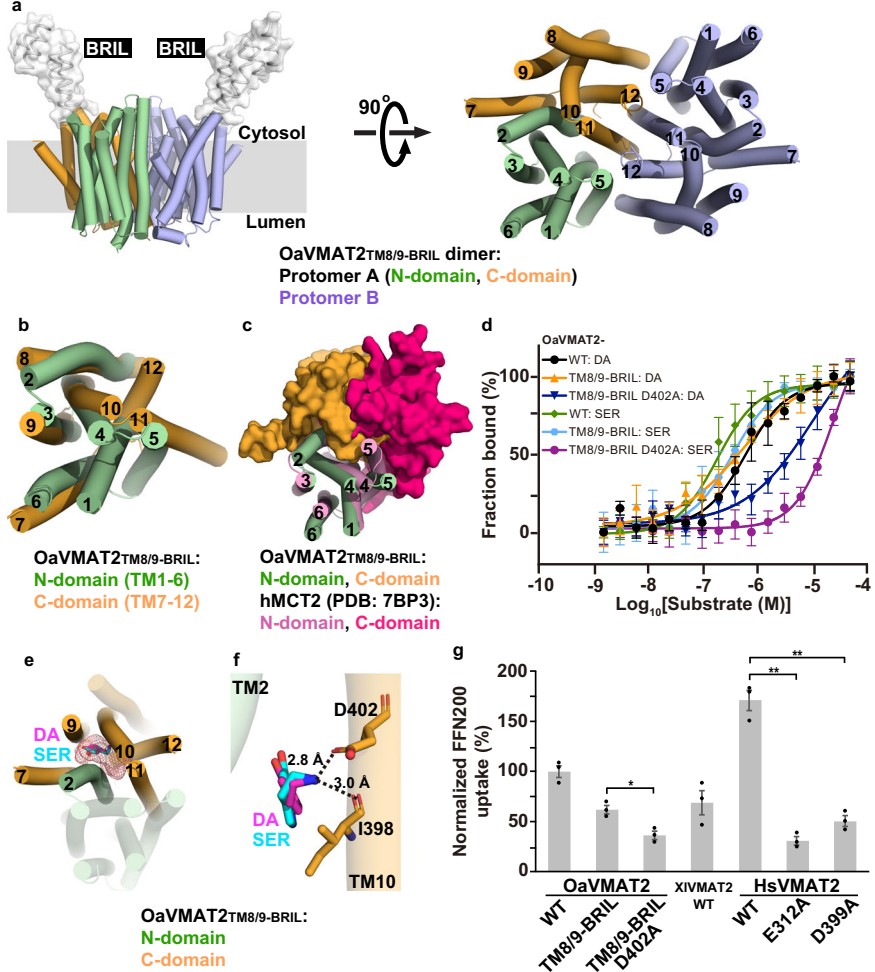

**Fig. 1 | Structure of OaVMAT2$_{TM8/9-BRIL}$. a** Cryo-EM structure of OaVMAT2$_{TM8/9-BRIL}$ dimer. Left, OaVMAT2$_{TM8/9-BRIL}$ dimer viewed parallel to the membrane (gray rectangle). N- and C-domains of protomer A are colored in green and orange, respectively, while protomer B in light blue. BRIL modules are displayed as white surface. Right, OaVMAT2$_{TM8/9-BRIL}$ dimer viewed perpendicular to the membrane from the cytosolic side. Twelve TM helices of VMAT2 are labeled with numbers throughout the manuscript. BRIL modules are omitted for clearer views for the rest of the manuscript. **b** Superposition of OaVMAT2$_{TM8/9-BRIL}$ C-domain (in orange) onto OaVMAT2$_{TM8/9-BRIL}$ N-domain (in green). **c** Superposition of OaVMAT2$_{TM8/9-BRIL}$ protomer onto hMCT2 protomer (PDB: 7BP3) by aligning N-domains (green cartoon for OaVMAT2$_{TM8/9-BRIL}$ and pink cartoon for hMCT2). C-domains are displayed as orange surface for OaVMAT2$_{TM8/9-BRIL}$ and dark pink surface for hMCT2. **d** MST fitting curves of DA/SER binding to OaVMAT2 WT and variants as indicated, N = 3 repeats with biologically independent protein samples.

Data are presented as mean ± SEM. Source data are provided as a Source Data file. **e** Docking of DA (magenta sticks) and SER (cyan sticks) in OaVMAT2$_{TM8/9-BRIL}$. The cavity between N-domain (green cartoon) and C-domain (orange cartoon) is indicated by red mesh. **f** Potential polar interactions between docked DA (magenta sticks) or SER (cyan sticks) and OaVMAT2$_{TM8/9-BRIL}$ (green and orange cartoons) are indicated by black dashed lines. Participating residues are displayed as orange sticks. **g** Normalized FFN200 uptake by HEK293T cells expressing various VMAT2 WT and variants as indicated, N = 3 experiments with biologically independent cells cultured in different wells. Data are presented as mean ± SEM. Source data are provided as a Source Data file. Individual data points are shown as black dots. Two-tailed Student's t-test was performed between the "TM8/9-BRIL" group and the "TM8/9-BRIL D402A" group (P = 0.012) for OaVMAT2, and between "WT" group and "E312A" group (P = 0.00022) or "D399A" group (P = 0.00049) for HsVMAT2. *P < 0.05; **P < 0.001.

(Supplementary Fig. 4d). Therefore, the unique organization of OaVMAT2$_{TM8/9-BRIL}$ C-domain breaks the two-fold pseudosymmetry between TM7-9 and TM10-12 (Supplementary Fig. 4e), as well as the pseudosymmetry between N- and C-domains (Fig. 1b) usually seen in a canonical MFS fold[7,8]. As a result, OaVMAT2$_{TM8/9-BRIL}$ has a different N-C interface than that of canonical MFS transporters (Fig. 1c). Furthermore, the OaVMAT2$_{TM8/9-BRIL}$ dimer interface is also distinct from that of hMCT2[12], which involves four TM helices (TM1, TM5, TM6 and TM8) (Supplementary Fig. 4f). Instead, the OaVMAT2$_{TM8/9-BRIL}$ dimer interface is formed by a different set of TM helices (TM5, TM8, TM11 and TM12) of each protomer (Fig. 1a), which buries an extensive interface area of 3980 Å².

It is curious that the OaVMAT2$_{TM8/9-BRIL}$ structure folds in a different way than a canonical MFS fold. Is this fold common to VMAT2? Or is it unique to OaVMAT2$_{TM8/9-BRIL}$ due to the protein engineering (i.e. replacement of the TM8/9 loop with BRIL)? Although a compact structure and an extensive dimer interface suggests that the OaVMAT2$_{TM8/9-BRIL}$ structure is well-organized, functional experiments were performed to validate this structure. First, both purified OaVMAT2$_{WT}$ and OaVMAT2$_{TM8/9-BRIL}$ samples were tested for substrate binding. Dopamine and serotonin bind to OaVMAT2$_{WT}$ with equilibrium dissociation constants ($K_d$) of $0.87 \pm 0.55$ μM and $0.21 \pm 0.06$ μM (Fig. 1d and Supplementary Table 2), respectively, comparable to previously reported values for VMAT2[13,14]. Interestingly, OaVMAT2$_{TM8/9-BRIL}$ retains binding capability to both dopamine ($K_d = 0.68 \pm 0.23$ μM) and serotonin ($K_d = 0.33 \pm 0.10$ μM) (Fig. 1d and Supplementary Table 2). Structural analysis revealed an occluded, empty cavity between N-domain and C-domain in each OaVMAT2$_{TM8/9-BRIL}$ protomer, which is formed between TM2, TM7, and TM9-12 (Fig. 1e). The cavity is sufficiently large to accommodate monoamine substrates such as dopamine or serotonin, which were subsequently docked into the cavity (Fig. 1e). Top-scored docking poses of dopamine and serotonin were placed similarly between TM2 and TM10, forming hydrophobic contacts with side chains of L147, P150, I398 and V401 (Supplementary Fig. 4g). For clarity, residue numbering of OaVMAT2$_{TM8/9-BRIL}$ is based on the wild-type OaVMAT2, not including BRIL. Notably, the basic amine group of dopamine/serotonin interacts with the acidic side group of D402 (TM10) via a salt bridge and the main-chain carbonyl oxygen of I398 (TM10) via a hydrogen bond (Fig. 1f). Besides, hydroxyl groups of dopamine/serotonin also form potential hydrogen bonds with the main-chain carbonyl oxygen of L147 (TM2) (Supplementary Fig. 4h). Supportively, a D402A mutation of OaVMAT2$_{TM8/9-BRIL}$ significantly weakened its binding to dopamine ($K_d = 11.2 \pm 1.8$ μM) and serotonin ($K_d = 20.0 \pm 2.7$ μM) (Fig. 1d and Supplementary Table 2). Meanwhile, the VMAT2 inhibitor reserpine bound to OaVMAT2$_{WT}$ with a high affinity ($K_d = 16.0 \pm 5.7$ nM), but did not bind to OaVMAT2$_{TM8/9-BRIL}$ (Supplementary Fig. 4i and Supplementary Table 2). This data is consistent with that the cavity in OaVMAT2$_{TM8/9-BRIL}$ is not large enough to accommodate reserpine (Fig. 1e), which also suggests that OaVMAT2$_{TM8/9-BRIL}$ folds differently than OaVMAT2$_{WT}$.

To investigate if the non-canonical OaVMAT2$_{TM8/9-BRIL}$ structure was a misfolded artifact, we further tested transport activities of OaVMAT2$_{WT}$ and OaVMAT2$_{TM8/9-BRIL}$ using a cell-based uptake assay. In the assay, a selective fluorescent substrate for VMAT2, FFN200 (fluorescent false neurotransmitter 200)[15], was transported into HEK293T cells that express OaVMAT2$_{WT}$ or OaVMAT2$_{TM8/9-BRIL}$, and the fluorescence accumulated in cells was subsequently measured (see Methods for details). Interestingly, while OaVMAT2$_{WT}$-expressing cells displayed a robust uptake activity for FFN200, OaVMAT2$_{TM8/9-BRIL}$-expressing cells also accumulated ~62% of FFN200 fluorescence compared to OaVMAT2$_{WT}$-expressing cells (Fig. 1g and Supplementary Table 3). Furthermore, accumulation of FFN200 fluorescence decreased substantially in cells expressing the OaVMAT2$_{TM8/9-BRIL}$ D402A mutant compared to OaVMAT2$_{TM8/9-BRIL}$-expressing cells (Fig. 1g and Supplementary Table 3). These data suggest that

OaVMAT2$_{TM8/9-BRIL}$ is functional (or partially functional) in both substrate binding and transport, and residue D402 contributes to both substrate binding and transport in OaVMAT2$_{TM8/9-BRIL}$. Certainly, evaluation of the biological relevance of this non-canonical fold of OaVMAT2$_{TM8/9-BRIL}$ requires further investigation. However, the finding of OaVMAT2$_{TM8/9-BRIL}$ with substrate binding and transport activities underscores that extra caution may be necessary when interpreting structures of engineered proteins.

## Cryo-EM structure of wild-type XlVMAT2$_{WT}$

We suspect that the non-canonical fold of OaVMAT2$_{TM8/9-BRIL}$ may be caused by the engineering of BRIL in the TM8/9 loop region of OaVMAT2 (Supplementary Fig. 1b). Therefore, we turned back to wild-type HsVMAT2$_{WT}$, OaVMAT2$_{WT}$ and XlVMAT2$_{WT}$ (Supplementary Fig. 1a) for more cryo-EM single-particle analysis. Eventually, the structure of XlVMAT2$_{WT}$ was determined at 4.0 Å resolution (Supplementary Fig. 5 and Table 1) after extensive data collection and processing. A total of 348 out of 514 residues were traced in the XlVMAT2$_{WT}$ structure, including all 12 TM helices and several connecting loops (Fig. 2a and Supplementary Fig. 6). Intriguingly, comparison to the hMCT2 structure revealed that the XlVMAT2$_{WT}$ structure adopts a canonical MFS fold (Supplementary Fig. 7a). Unlike OaVMAT2$_{TM8/9-BRIL}$, N- and C-domains of XlVMAT2$_{WT}$ are structurally similar, and could be superimposed with a C$_\alpha$ RMSD of 2.5 Å (Fig. 2b). Comparison between the OaVMAT2$_{TM8/9-BRIL}$ and XlVMAT2$_{WT}$ structures revealed that their N-domains align relatively well (C$_\alpha$ RMSD of 1.4 Å), but their C-domains differ substantially (C$_\alpha$ RMSD of 10.3 Å) in the arrangement of TM7 and TM10-12 (Supplementary Fig. 7b). Meanwhile, purified XlVMAT2$_{WT}$ showed reasonable binding affinities for substrates dopamine ($K_d = 1.13 \pm 0.52$ μM) and serotonin ($K_d = 0.43 \pm 0.22$ μM), and for the inhibitor reserpine ($K_d = 17.0 \pm 8.8$ nM) (Fig. 2c and Supplementary Table 2). Furthermore, cells expressing XlVMAT2$_{WT}$ also exhibited a reasonable uptake activity for FFN200 (Fig. 1g and Supplementary Table 3). These data suggest that the XlVMAT2$_{WT}$ structure likely represents a canonical VMAT2 fold.

The XlVMAT2$_{WT}$ structure was solved in an apo, lumen-facing open conformation with a large, deep cavity (Fig. 2d). A cytosolic gate[16] formed by M203 (TM4), M220 (TM5), M402 (TM10) and Y421 (TM11) seals the cytosolic side of the transporter, whereas the luminal opening allows luminal solvent to reach deeply to the cytosolic gate (Supplementary Fig. 7c). A substrate (serotonin) was then docked into the cavity of XlVMAT2$_{WT}$. Serotonin was placed in a pocket between TM5, TM7, TM8, TM10 and TM11 near the center of XlVMAT2$_{WT}$ (Fig. 2d). The hydrophobic moiety of serotonin interacts with L227 (TM5), V231 (TM5), Y340 (TM8), F428 (TM11) and F432 (TM11) through hydrophobic interactions (Fig. 2e). Meanwhile, the hydroxyl group of serotonin forms a hydrogen bond with the side chain of E311 (TM7), and its amine group interacts with side chains of N304 (TM7) and D398 (TM10) via a hydrogen bond and a salt bridge, respectively (Fig. 2f). Supportively, some of these residues (e.g. the equivalent residues of E311 and D398 in other species) have been shown previously to be critical for substrate transport in VMAT2 in modeling and functional studies[14,17]. Furthermore, the protein sequence of VMAT2 is highly conserved among different species (Supplementary Fig. 8). For example, XlVMAT2 and HsVMAT2 share ~80% sequence identity and ~94% homology, which contain nearly identical residues that form the aforementioned serotonin binding site (Supplementary Fig. 8).

To validate the substrate binding model in VMAT2, the corresponding serotonin site in HsVMAT2 was mutated and tested for serotonin binding. Supportively, all tested mutants of HsVMAT2 serotonin site, including L228A, V232A, E312A, Y341A, D399A and Y433A, reduced their binding affinities for serotonin (Fig. 2g and Supplementary Table 2). Among them, HsVMAT2 D399A (D398 in XlVMAT2) and E312A (E311 in XlVMAT2), which are predicted to form polar interactions with serotonin before mutation (Fig. 2f), had the most significant influence on serotonin binding (Fig. 2g and Supplementary Table 2). Meanwhile,

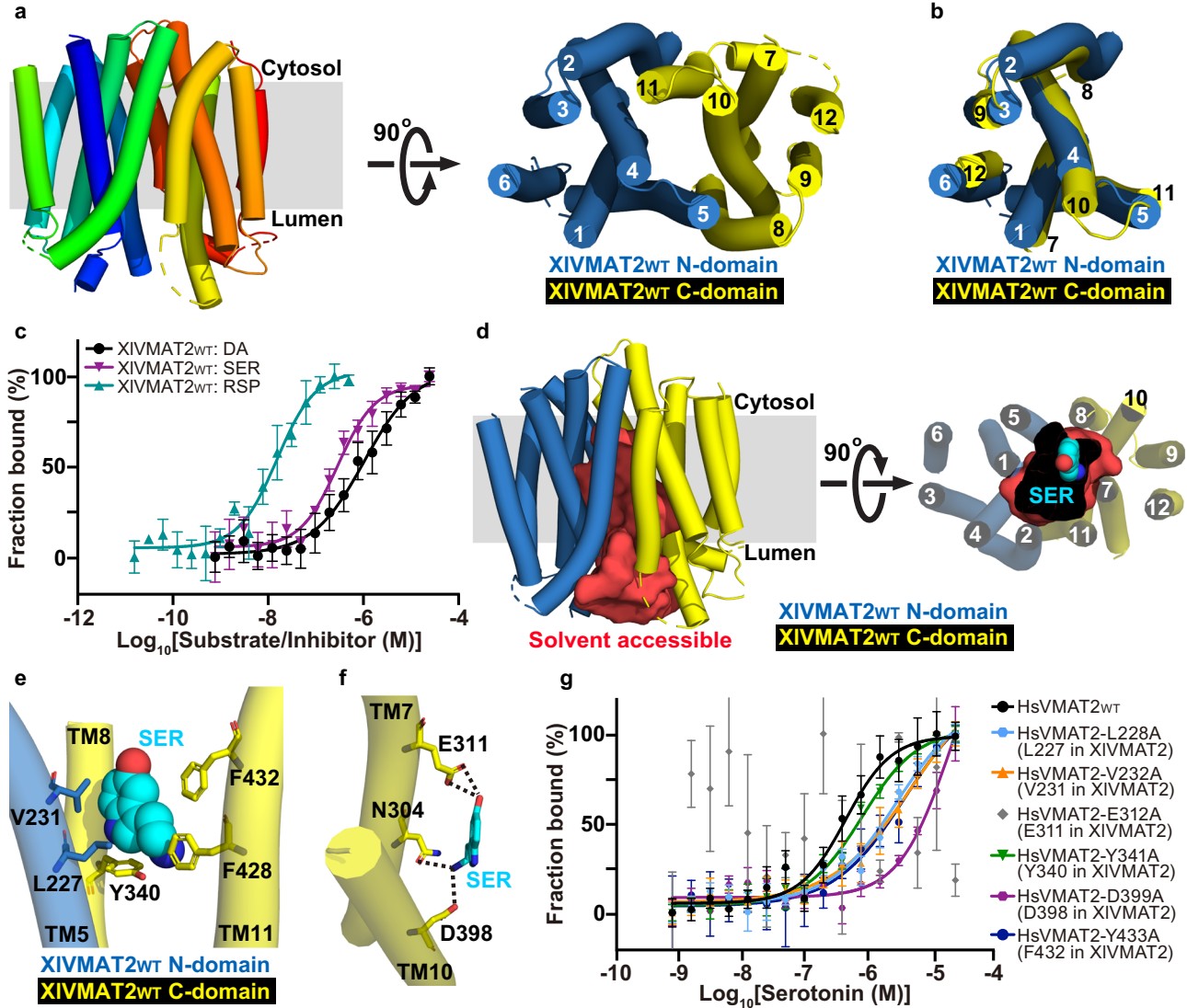

**Fig. 2 | Structure of XlVMAT2$_{WT}$. a** Cryo-EM structure of XlVMAT2$_{WT}$. Left, XlVMAT2$_{WT}$ (in spectrum color) viewed parallel to the membrane (gray rectangle). Right, XlVMAT2$_{WT}$ viewed perpendicular to the membrane from the cytosolic side. N- and C-domains are colored in blue and yellow, respectively. **b** Superposition of XlVMAT2$_{WT}$ C-domain (in yellow) onto XlVMAT2$_{WT}$ N-domain (in blue). **c** MST fitting curves of DA/SER/RSP binding to XlVMAT2$_{WT}$, $N = 3$ repeats with biologically independent protein samples. Data are presented as mean ± SEM. Source data are provided as a Source Data file. **d** Solvent accessibility analysis and docking of SER in XlVMAT2$_{WT}$ (blue and yellow cartoons). The solvent-accessible space is displayed

as red surface. Left, viewed parallel to the membrane. Right, viewed perpendicular to the membrane from the luminal side. Docked SER is displayed as cyan spheres. **e** Potential hydrophobic interactions between docked SER (cyan spheres) and XlVMAT2$_{WT}$ (blue and yellow cartoons). Participating residues are shown as sticks. **f** Potential polar interactions between docked SER (cyan sticks) and XlVMAT2$_{WT}$ (yellow cartoon) are indicated by black dashed lines. Participating residues are shown as yellow sticks. **g** MST fitting curves of SER binding to HsVMAT2 WT and variants as indicated, $N = 3$ repeats with biologically independent protein samples. Data are presented as mean ± SEM. Source data are provided as a Source Data file.

cells expressing HsVMAT2$_{WT}$ showed a robust uptake activity for FFN200, which was significantly reduced when HsVMAT2 harbors the D399A or E312A mutation (Fig. 1g and Supplementary Table 3). These data are consistent with the docking model of serotonin in XlVMAT2$_{WT}$.

## Discussion

One interesting point worth discussing is the oligomeric state of our VMAT2 structures. The non-canonical structure of OaVMAT2$_{TM8/9-BRIL}$ forms a dimer (Fig. 1a), whereas the canonical structure of XlVMAT2$_{WT}$ is a monomer (Fig. 2a). In OaVMAT2$_{TM8/9-BRIL}$, the dimer interface is formed mainly by residues from TM5 (A232, L236, S243, V244), TM8 (W331, L338), TM11 (V418, Y421, Y425, A428, D429, F432) and TM12 (I461, I464, L465, P468, L469, F471) (Supplementary Fig. 9a). These residues are highly conserved among different species (Supplementary Fig. 8). However, in a canonical MFS fold (e.g. XlVMAT2$_{WT}$), many

corresponding residues (e.g. V414, Y417, Y421, A424, D425, F428 of TM11 in XlVMAT2) are facing the interior of VMAT2 and are not available for dimerization (Supplementary Fig. 9b). This result may explain why OaVMAT2$_{TM8/9-BRIL}$ forms a dimer but XlVMAT2$_{WT}$ does not. To evaluate the biological relevance of OaVMAT2$_{TM8/9-BRIL}$ dimer, we further compared its oligomeric state to that of wild-type OaVMAT2$_{WT}$ by a chemical crosslinking assay. Consistently, OaVMAT2$_{TM8/9-BRIL}$ in the detergent solution was readily crosslinked and showed a dimer band (Supplementary Fig. 1e). Meanwhile, OaVMAT2$_{WT}$ showed a monomer band only and no dimer band after crosslinking (Supplementary Fig. 1e). Interestingly, crosslinked OaVMAT2$_{TM8/9-AmpC}$ also showed a minor dimer band, while the rest of the protein appeared as a monomer band (and a high-molecular-weight aggregate band) (Supplementary Fig. 1e). This data suggests that the oligomeric state of wild-type OaVMAT2$_{WT}$ is different from that of OaVMAT2$_{TM8/9-BRIL}$

(and maybe also OaVMAT2$_{TM8/9\text{-}AmpC}$), and the dimer form of OaVMAT2$_{TM8/9\text{-}BRIL}$ may be an artifact induced by its non-canonical fold and engineering.

Another point worth mentioning is the effect of pH on VMAT2 conformations. VMAT2 is a H$^+$-driven transporter and pH may affect its conformation[1,2]. Supportively, purified XlVMAT2 protein showed distinct enzymatic digestion patterns at pH 6.0 (mimicking the vesicular lumen pH) vs. pH 7.5 (mimicking the cytosol pH) (Supplementary Fig. 9c), suggesting that XlVMAT2 likely assumes different conformations at these two pHs. Therefore, the XlVMAT2 sample was actually prepared in both pH 6.0 (MES) and pH 7.5 (Tris) buffers for cryo-EM data collection, in an attempt to solve two different conformations with these two pH conditions. Unfortunately, only the pH 6.0 data set allowed a structure solution at ~4 Å after extensive data processing (Table 1). On the other hand, the OaVMAT2$_{TM8/9\text{-}BRIL}$ structure (pH 7.5) and the XlVMAT2$_{WT}$ structure (pH 6.0) adopt different conformations (different folds) (Supplementary Fig. 7b). However, their difference is more likely attributed to the fused BRIL in OaVMAT2$_{TM8/9\text{-}BRIL}$ rather than different pHs, thus not suitable for discussing the pH effect on VMAT2 conformations.

During preparation of this manuscript, several papers on cryo-EM structures of human VMAT2 were published[18–21]. Therefore, we compared those structures to our structures. Human VMAT2 structures adopt a canonical MFS fold and are different from the OaVMAT2$_{TM8/9\text{-}BRIL}$ structure (Supplementary Fig. 10a). Meanwhile, the XlVMAT2$_{WT}$ structure is highly similar to a serotonin-bound, lumen-facing structure of human VMAT2 (PDB: 8JSW) reported by Wu and colleagues[20] (Supplementary Fig. 10b). Interestingly, our docked serotonin model in XlVMAT2$_{WT}$ is very close to the serotonin molecule in the human VMAT2-serotonin structure[20] (PDB: 8JSW) (Supplementary Fig. 10c). This result supports the validity of the XlVMAT2$_{WT}$ structure and the substrate binding model proposed by molecular docking. It is noteworthy that unlike all human VMAT2 cryo-EM studies mentioned here that have fused a binding-pair to both N- and C-termini of the transporter to restrict terminal flexibility[18–21], XlVMAT2$_{WT}$ consists of completely native sequence. Besides, given the curious case of OaVMAT2$_{TM8/9\text{-}BRIL}$, which shows a non-canonically folded MFS transporter with substrate binding and transport activity, extra caution is advised when interpreting engineered membrane protein structures.

## Methods

### Protein expression and purification

The genes encoding full-length wild-type VMAT2s from *Homo sapiens* (HsVMAT2, NCBI accession: NP_003045.2), *Ovis aries* (OaVMAT2, NCBI accession: XP_004020268.1), *Xenopus laevis* (XlVMAT2, NCBI accession: XP_018080488.1) and other 9 species were synthesized (Genewiz, China) and cloned into a modified pPICZ plasmid (Thermo Fisher Scientific) containing an amino-terminal decahistidine tag, followed by the bacterial cytochrome b562RIL (BRIL)[22] and tobacco etch virus (TEV) protease recognition site. In addition, a TEV recognition sequence-BRIL-decahistidine tag was fused to the carboxy-terminus of VMAT2. For VMAT2 engineering, either BRIL or a β-lactamase AmpC from *Pseudomonas aeruginosa*[23] was inserted to replace various inter-helix loop regions of VMAT2 (Supplementary Table 1). Point mutations were introduced by site-directed mutagenesis using QuikChange II system (Agilent) according to manufacturer's recommendation, and were verified by sequencing. Wild-type VMAT2 and variants were overexpressed in yeast (*Pichia pastoris* strain GS115) cells by adding 1% (v/v) methanol and 2.5% (v/v) dimethyl sulfoxide (DMSO) at an optical density (OD) of ~2 at 600 nm and shaking at 30 °C for 48 h. Cell pellets were resuspended in lysis solution (LS) containing 20 mM Tris-HCl pH 7.5, 150 mM NaCl, 10% (v/v) glycerol, 1 mM phenylmethanesulfonyl fluoride (PMSF) and 2 mM β-mercaptoethanol, and were lysed by an AH-1500 high-pressure homogenizer (ATS, China) at 1,300 MPa. Undisrupted cells and cell debris were separated by centrifugation at 3,000 x g for 10 min, and membranes were collected by ultracentrifugation at 140,000 x g for 1 h at 4 °C. Protein was extracted by addition of 1% (w/v) n-dodecyl-β-D-maltopyranoside (DDM, Anatrace) at 4 °C for 2 h and the extraction mixture was centrifuged at 200,000 x g for 20 min at 4 °C. The supernatant was then loaded onto a cobalt metal affinity column and was washed with 20 bed-volume of LS containing 1 mM DDM and 56 mM imidazole pH 8.0. VMAT2 protein was released from the cobalt column by TEV protease cleavage at room temperature for 2 h.

### Cryo-EM sample preparation and image acquisition

Affinity-purified VMAT2s were concentrated to 10-16 mg/ml and loaded onto a Superdex 200 Increase 10/300 GL column (Cytiva), and were further purified by size-exclusion chromatography (SEC). For the OaVMAT2$_{TM8/9\text{-}BRIL}$ structure, the SEC buffer contained 20 mM Tris-HCl pH 7.5, 150 mM NaCl, 5 mM β-mercaptoethanol, and 0.4 mM DDM. For the XlVMAT2$_{WT}$ structure, the SEC buffer contained 20 mM MES-Na pH 6.0, and 150 mM NaCl, 5 mM β-mercaptoethanol, and 0.4 mM DDM. For those unsuccessful reconstructions, their details are omitted in this manuscript. For cryo-EM grid preparation, freshly SEC-purified VMAT2 was concentrated to ~8 mg/ml and 3.0 μl of protein solution was applied to glow-discharged (60 s) holey carbon grids (Quantifoil

**Table 1 | Cryo-EM data collection and model statistics**

| | OaVMAT2$_{TM8/9\text{-}BRIL}$ (EMD-38389) (PDB: 8XIT) | XlVMAT2$_{WT}$ (EMD-38390) (PDB: 8XIU) |
|---|---|---|
| **Data collection and processing** | | |
| Movies | 6149 | 8192 |
| Magnification | 165,000× | 165,000× |
| Voltage (kV) | 300 | 300 |
| Electron exposure (e$^-$/Å$^2$) | 62.64 | 59.1 |
| Defocus range (μm) | −1.0 to −1.6 | −1.1 to −1.6 |
| Pixel size (Å) | 0.85 | 0.85 |
| Symmetry imposed | C2 | - |
| Initial particle images (no.) | 1,064,611 | 1,272,193 |
| Final particle images (no.) | 304,899 | 107,961 |
| Map resolution (Å) | 3.2 | 4.0 |
| FSC threshold | 0.143 | 0.143 |
| **Refinement** | | |
| **Model composition** | | |
| Non-hydrogen atoms | 6002 | 2615 |
| Protein residues | 784 | 348 |
| Ligands | - | - |
| **R.m.s. deviations** | | |
| Bond lengths (Å) | 0.003 | 0.002 |
| Bond angles (°) | 0.644 | 0.554 |
| **Validation** | | |
| MolProbity score | 1.22 | 1.35 |
| Clashscore | 4.42 | 5.99 |
| Rotamer outliers (%) | 0.00 | 0.00 |
| **Ramachandran plot** | | |
| Favored (%) | 98.02 | 98.51 |
| Allowed (%) | 1.98 | 1.49 |
| Disallowed (%) | 0.00 | 0.00 |
| *B*-factors (Å$^2$) | | |
| Protein | 80.42 | 139.87 |
| Ligand | - | - |

Au R2/1, 200 mesh). The grids were blotted with filter paper for 3.0 s at 4 °C and 100% humidity and frozen by plunging into liquid ethane, which had been cooled with liquid nitrogen, using a Vitrobot Mark IV system (Thermo Fisher Scientific). The frozen grids were loaded into a Titan Krios cryogenic electron microscope (Thermo Fisher Scientific) operated at 300 kV with the condenser lens aperture set to 50 μm. Microscope magnification was at 165,000×, corresponding to a calibrated sampling of 0.85 Å per physical pixel. Movie stacks were collected automatically using the EPU software (Version 2.9.0.1519REL, Thermo Fisher Scientific) on a K2 Summit direct electron device (Gatan) equipped with a BioQuantum GIF energy filter operated at 20 eV in counting mode. The recording rate was set to 5 raw frames per second, and the total exposure time was set to 6 s, yielding 30 frames per stack and a total dose of ~60 e$^-$/Å$^2$. Two sets of movie stacks (3183 + 2966 = 6149) were collected for OaVMAT2$_{TM8/9\text{-BRIL}}$, and a total of 8192 movie stacks were collected for XlVMAT2$_{WT}$. The defocus range during data collection was set from −1.0 to −1.6 μm.

## Cryo-EM data processing and 3D reconstruction

Cryo-EM data processing was performed using EMAN[24] (v2.2), Relion[25] (v4.0) and cryoSPARC[26] (v4.3). For OaVMAT2$_{TM8/9\text{-BRIL}}$ set 1, a total of 3183 movie stacks were motion-corrected and dose-weighted using MotionCor2[27]. Contrast transfer function (CTF) was estimated using CTFFIND[28] (v4.1). Micrographs were visually inspected to remove poor-quality images, and the resulting 3,060 micrographs were subjected to neural network particle picking by EMAN. A total of 1,086,368 particles were extracted in Relion using a box size of 256 pixels. Extracted particles were downscaled two-fold and were subjected to reference-free 2D classification. After two rounds of 2D classification, 362,481 particles in the best 2D classes were selected, re-extracted and imported into cryoSPARC. Then after two rounds (5-class and 3-class) of ab-initio reconstruction and heterogeneous refinement, 161,705 particles of the best 3D class were subjected to non-uniform[29] and local refinements, yielding a 4.41 Å density map (Supplementary Fig. 2). For OaVMAT2$_{TM8/9\text{-BRIL}}$ set 2, similar motion and CTF corrections were applied to 2966 movie stacks and 2806 micrographs were selected after visual examination. Using 2D templates from OaVMAT2$_{TM8/9\text{-BRIL}}$ set 1, a total of 1,105,661 particles were auto-picked and extracted from OaVMAT2$_{TM8/9\text{-BRIL}}$ set 2 by cryoSPARC, among which 702,130 particles were selected after 2D classification. After one round of 6-class ab-initio reconstruction and heterogeneous refinement, 304,729 particles of the best 3D class were subjected to non-uniform and local refinements, yielding a 3.57 Å density map (Supplementary Fig. 2). By merging particles corresponding to the 4.41 Å and 3.57 Å maps, a total of 466,434 particles were subjected to one round of 5-class ab-initio reconstruction and heterogeneous refinement, and 304,899 particles of the best 3D class were imported into Relion for auto-refinement and Bayesian polishing. Polished particles were imported back into cryoSPARC for non-uniform and local refinements, yielding a final density map of 3.2 Å for OaVMAT2$_{TM8/9\text{-BRIL}}$ (Supplementary Fig. 2).

For XlVMAT2$_{WT}$, 8192 movies were motion-corrected by Motion-Cor2 and CTF-estimated by Gctf[30]. Images with a CTF resolution better than 6 Å were selected, yielding 7433 images. A subset of the data (500 images) were used to auto-pick particles by the Blob Picker function in cryoSPARC. After 2D classification, 52,341 particles with different views were selected to train the deep picker program Topaz[31] (v0.2.4). After training, a total of 1,272,193 particles were picked and extracted by Topaz with a box size of 256 pixels. Extracted particles were binned by a factor of 2 for 2D classification, ab-initio reconstruction, and heterogeneous refinement in cryoSPARC. 282,506 particles of the best 3D class were re-extracted with a box size of 256 pixels, which yielded a 4.8 Å density map after non-uniform refinement with customized settings (Supplementary Fig. 5). To further improve the density map, seed-facilitated guided multi-reference 3D classification[32] was performed in cryoSPARC via heterogeneous refinement to generate the

best 3D class. The multi-references include: (1) accurate and biased references generated by ab-initio reconstruction; (2) resolution gradient maps (the 4.8 Å map and two low-pass filtered maps at 12 Å and 18 Å); (3) noise re-weighted maps (two de-noised maps whose micelle portion was downscaled by a factor of 0.3 and 0.7, and an empty micelle map). After removal of duplicate particles, 435,605 particles were re-extracted and subjected to ab-initio reconstruction, heterogeneous refinement (with noise re-weighted maps), and non-uniform refinement, resulting in a 4.5 Å density map containing 286,898 particles (Supplementary Fig. 5). The particles were imported into Relion for 3D auto-refinement with a tight TM mask (1.8° local angular search), followed by Bayesian polishing. Then a 4-class ($K = 4$) skip-alignment 3D refinement was performed and 107,961 particles of the best 3D class were selected. Polished particles were imported back into cryoSPARC for non-uniform refinement, and several rounds of CTF refinement and local refinement, yielding a final density map of 4.0 Å for XlVMAT2$_{WT}$ (Supplementary Fig. 5). The final map resolutions for both OaVMAT2$_{TM8/9\text{-BRIL}}$ and XlVMAT2$_{WT}$ were estimated using the 0.143 criterion of the Fourier shell correlation (FSC) curve in cryoSPARC.

## Model building and structure refinement

Model building of OaVMAT2$_{TM8/9\text{-BRIL}}$ and XlVMAT2$_{WT}$ were facilitated by an AlphaFold[33]-predicted human VMAT2 model (AlphaFold DB: AF-Q05940-F1). Both OaVMAT2$_{TM8/9\text{-BRIL}}$ and XlVMAT2$_{WT}$ models were manually adjusted and rebuilt in Coot (v0.9.3 EL)[34] and refined using the phenix.real_space_refine[35] module in PHENIX (v1.19.1-4122)[36]. Some flexible inter-helix loop regions were removed due to unresolved densities. Model geometries were assessed by Molprobity[37] and summarized in Table 1. All structural figures, RMSD calculations, length and surface area measurements were performed in PyMOL (v2.5.4) (Schrödinger, LLC). Some cryo-EM densities were visualized in UCSF Chimera (v1.15)[38]. Accessibility analysis was performed using a volume-filling program HOLLOW (v1.1)[39] with default settings.

## Molecular docking and energy minimization

Docking analysis was performed using AutoDock Vina[40] for modeling dopamine (DA) and serotonin (SER) into the OaVMAT2$_{TM8/9\text{-BRIL}}$ structure, and SER into the XlVMAT2$_{WT}$ structure. Protein models were prepared according to AutoDock Vina manual, and polar hydrogens were added using AutoDock Tools[41]. DA and SER coordinates were generated by UCSF Chimera[38] from SMILES strings, and were prepared according to AutoDock Vina manual by merging non-polar hydrogens and verifying rotatable bonds in AutoDock Tools. The docking grid was set to encompass the protein central cavity with the aid of model visualization in AutoDock Tools, and docking trials were performed with high exhaustiveness. Top-scored docking models with lowest binding energies were manually examined and further optimized by the AMMOS2 web server[42] (http://drugmod.rpbs.univ-paris-diderot.fr/ammosHome.php). AMMOS2 employs an automatic procedure for energy minimization of protein-ligand complexes. For each docked model, the input files for the protein (OaVMAT2$_{TM8/9\text{-BRIL}}$ or XlVMAT2$_{WT}$, as a.pdb file) and the ligand (DA or SER, as a.mol2 file) were prepared according to the AMMOS2 user guide. Energy minimization was performed with AMMOS2 default settings.

## Microscale thermophoresis (MST)

Binding of DA/SER/RSP to VMAT2s was analyzed by MST. MST analysis was performed using Monolith NT.115 (MO.Control software v1.6.1) (NanoTemper, Germany) by staining His-tagged VMAT2s with the RED-Tris-NTA 2nd Generation dye (NanoTemper, Germany). Affinity-purified VMAT2s were further purified by SEC in a buffer containing 20 mM MES-Na pH 6.0, 150 mM NaCl, and 0.4 mM DDM. Peak fractions were pooled and diluted to 200 nM using the SEC buffer. Protein samples were mixed with 100 nM RED-Tris-NTA 2nd Generation dye at a 1:1 ratio and incubated for 30 min at room temperature. Then the

sample was centrifuged at 15,000 × g for 10 min at 4 °C to keep the supernatant containing the labeled protein. A diluted series of DA/SER/RSP were prepared as per the MST manual. DA and SER were prepared as 100 µM stock solutions in the SEC buffer. RSP was solubilized in methanol at 10 µM, which was further diluted with the SEC buffer to 1 µM as the stock solution. The highest concentration of titrants was 100 µM for DA/SER and 1 µM for RSP. Labeled VMAT2 was mixed with serial-diluted DA/SER/RSP and incubated for 30 min at room temperature. Then samples were loaded into capillaries and MST measurements were performed according to the Monolith manual. Fluorescent signals of the bound fraction were normalized in a 0-100% scale. The equilibrium dissociation constant ($K_d$) was determined using the MO.Affinity Analysis software (v2.2.4) (NanoTemper, Germany) with the $K_d$ fit function. A noise-like pattern for all data points in a measurement that fails the $K_d$ fit function was deemed no-binding. All MST measurements were performed in three biologically independent experiments ($N = 3$ biologically independent protein samples). The $K_d$ values are listed in Supplementary Table 2, and are expressed as mean ± standard error of the mean (SEM) in the text and figures. Two-tailed Student's t-test was performed for statistical analysis in Supplementary Table 2.

### FFN200 uptake assay

Genes encoding wild-type HsVAMT2, OaVMAT2, XlVMAT2 and their variants were cloned into a modified pCI mammalian expression vector (Promega) containing an amino-terminal FLAG tag (DYKDDDDK) and green fluorescence protein (GFP), followed by TEV protease recognition site and decahistidine. The recombinant plasmids were prepared using a plasmid maxiprep kit (QIAGEN) with final concentrations above 1 µg/µl. The FFN200 (fluorescent false neurotransmitter 200)[15] uptake assay was performed using a previously published procedure[21] with some modifications. Briefly, ~5 × 10^5 HEK293T cells (Procell; Catalog number: CL-0005) were plated in six-well plates and cultured in low glucose Dulbecco's modified Eagle medium (DMEM), 10% (v/v) fetal bovine serum (FBS) and 1% (w/v) penicillin/streptomycin at 37 °C with 5% $CO_2$. When cells reached 70–80% confluency, cells were transfected with various VMAT2 plasmids by replacing the original medium with 2 ml of low glucose DMEM containing a VMAT2 plasmid (4 µg) and 5 µl of PEI transfection reagent (MedChemExpress). After transfection for 36 h at 37 °C, 4 µl of 2.8 M glucose (final concentration of 5.6 mM) and 1 µl of 2 M $CaCl_2$ (final concentration of 1 mM) were added to the cell culture medium and cells were incubated at 37 °C for 30 min. Uptake was initiated by addition of 1 µl of 10 mM FFN200 (Tocris Bioscience) to the cell culture medium, followed by incubation for 1 h at 37 °C. The uptake of FFN200 was terminated by three washes of cold phosphate buffered saline (PBS, pH 7.4) supplemented with 0.2% (w/v) bovine serum albumin (BSA). Cells were then detached by treatment with 200 µl of 0.25% (w/v) trypsin-EDTA solution (in PBS) for 4 min at 37 °C, which was terminated by addition of 400 µl of cold PBS supplemented with 5% (v/v) FBS. The density of cells were counted by a Counterstar BioLab cell counter (Alit Biotech, China). For each sample, 10^6 cells were collected and lysed by sonication for four rounds of 5 min using a benchtop bath sonicator (Scientz Biotechnology, China) in the presence of 1% (w/v) DDM at 4 °C. The lysis solution was centrifuged at 20,000 x g for 1 h at 4 °C, and the supernatant containing FFN200 was collected for fluorescence measurement using a Duetta fluorescence spectrometer (HORIBA Scientific) with the excitation and emission wavelengths set at 352 nm and 451 nm, respectively. The accumulated FFN200 signal in each sample was calculated by subtracting the blank signal, which was obtained from cells transfected with a pCI-GFP plasmid. Calculated fluorescent signals were normalized in a 0–100% scale, using the signal from cells expressing wild-type OaVMAT2 as 100%. The FFN200 uptake assay was performed in three biologically independent experiments ($N = 3$ biologically independent cell samples). The absolute fluorescence values for each sample were reported in

Supplementary Table 3, and data are expressed as mean ± SEM. Two-tailed Student's t-test was performed for statistical analysis in Supplementary Table 3.

### Chemical crosslinking

Purified OaVMAT2$_{WT}$, OaVMAT2$_{TM8/9-BRIL}$ and OaVMAT2$_{TM8/9-AmpC}$ proteins (~10 µg each in 20 mM HEPES-Na pH 7.5, 150 mM NaCl, 5 mM β-mercaptoethanol, and 0.4 mM DDM) were crosslinked by glutaraldehyde (GA) at final concentrations of 4 mM, 2 mM and 1 mM for 10 min at room temperature. The crosslinking reaction was quenched by addition of 1 M Tris-HCl pH 7.5 to a final concentration of 20 mM, and the protein samples were subjected to SDS-PAGE analysis. Chemical crosslinking and SDS-PAGE experiments were repeated three times with similar results.

### Enzymatic digestion of XlVMAT2 protein

XlVMAT2 protein was purified by SEC in either 20 mM Tris-HCl pH 7.5 or 20 mM MES-Na pH 6.0 (two pH conditions), 150 mM NaCl, 5 mM β-mercaptoethanol, and 0.4 mM DDM. SEC-purified XlVMAT2 protein (~10 µg for each digestion) was treated with trypsin (TPCK-treated, Worthington Biochemical Corporation) or chymotrypsin (TLCK-treated, Worthington Biochemical Corporation) at three ratios of 10:1, 20:1 or 50:1 (w/w, XlVMAT2:enzyme) for 10 min at room temperature. The digestion reaction was terminated by addition of 100 mM PMSF to a final concentration of 1 mM, and the protein samples were subjected to SDS-PAGE analysis. Enzymatic digestion and SDS-PAGE experiments were repeated three times with similar results.

### Reporting summary

Further information on research design is available in the Nature Portfolio Reporting Summary linked to this article.

## Data availability

The data that support this study are available from the corresponding authors upon request. The cryo-EM density maps for OaVMAT2$_{TM8/9-BRIL}$ and XlVMAT2$_{WT}$ have been deposited in the Electron Microscopy Data Bank (EMDB) under the accession codes EMD-38389 (OaVMAT2$_{TM8/9-BRIL}$); and EMD-38390 (XlVMAT2$_{WT}$). The atomic coordinates of the OaVMAT2$_{TM8/9-BRIL}$ and XlVMAT2$_{WT}$ structures have been deposited in the Protein Data Bank (PDB) under the accession codes 8XIT (OaVMAT2$_{TM8/9-BRIL}$); and 8XIU (XlVMAT2$_{WT}$). Two previously published structures used in this study are available in the Protein Data Bank under accession codes 7BP3 (hMCT2 dimer) and 8JSW (hVMAT2-SER). The source data underlying Figs. 1d, g, 2c, g, Supplementary Fig. 1a, 1c, 1d and 4i, and Supplementary Table 2 and 3 are provided as a Source Data file. Uncropped gels of those presented in Supplementary Figs. (1a, 1c, 1d, 1e and 9c) are supplied at the end of the Supplementary Information file. Source data are provided with this paper.

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

## Acknowledgements

Cryo-EM data were collected at SKLB West China Cryo-EM Center and were processed at the Duyu High Performance Computing Center of Sichuan University. This work was supported in part by Sichuan Science and Technology Program grant 2023ZYD0125 to X.Z., the National Natural Science Foundation of China (NSFC) grants 31770783 to X.Z., 32222040 and 32070049 to Zh.S., West China Hospital of Sichuan University "1.3.5 Project" for Disciplines of Excellence grants ZYYC20014 to X.Z. and ZYYC21006 to Zh.S., and Sichuan University "From 0 To 1" Innovation Program grant 2023SCUH0067 to X.Z.

## Author contributions

X.Z. and Zi.S. conceived the project and wrote the manuscript with input from all authors. X.Z., Zi.S and Zh.S. designed the experiments. Y.L. and C.F. processed cryo-EM data, and performed structural, functional and computational studies; Zh.S. and H.M. collected and processed cryo-EM data; X.Z., Y.L., C.F. and H.M. built structural models; all authors analyzed the data.

## Competing interests

The authors declare no competing interests.
