## [Peer Review File · Nature Communications]

Engineering of a mammalian VMAT2 for cryo-EM analysis results in non-canonical protein foldingREVIEWER COMMENTS

Reviewer #1 (Remarks to the Author):

Key results

The authors solved the single particle cryo-EM structure of sheep VMAT2 with BRIL inserted into a loop to increase the molecular size and enhance the shape for structure determination. They obtained a high-quality cryo-EM reconstruction of the chimeric protein but found that the VMAT2 fold was significantly different than expected based on previous major facilitator superfamily (MFS) structures. The authors found that serotonin and dopamine bound the engineered protein with affinities similar to that of the wild-type protein. However, the VMAT2 inhibitor reserpine bound to the wild-type protein with high affinity but did not bind to the engineered protein, which confirmed that the unexpected fold was an artifact of the protein engineering. The authors succeeded in obtaining a 4-Angstrom cryo-EM map of apo, wild-type frog VMAT2 which revealed the expected MFS fold. The authors performed site-directed mutagenesis and associated serotonin and reserpine binding studies on both constructs and conducted computational docking with these molecules and their cryo-EM structures. During preparation of this manuscript, several structural studies on VMAT2 were published including Wu et al., Nature 2023 that reported four structures of an engineered human VMAT2 each bound either substrate or an inhibitor (Ref 17) and Pidathala et al. that reported structures an engineered human VMAT2 with inhibitor bound (Ref 16). The authors compared their wild-type, apo frog VMAT2 structure and computational docking results with the recently published VMAT2 structures, and they made the case that their work demonstrates that protein engineering to aid in structure determination may result in well-folded but physiologically irrelevant structures.

Validity

The authors performed careful cryo-EM structure determination and thorough structure-based mutagenesis and biochemical analysis. They used computational docking rather than experimentally determined structures to gain insight into substrate and inhibitor binding. They described similarities and differences between their docking results and related experimentally determined co-structures.

Significance

It is a good point that protein engineering for structure determination should be used with caution, and the successful generation of the ~4-Angstrom map of wild-type frog VMAT2 by single particle cryo-EM is technically impressive. However, the publications describing experimentally-determined apo, substrate-, and inhibitor-bound human VMAT2 structures that were noted and analyzed by the authors (Refs 15 – 18) greatly diminished the significance of the structural work presented here.

Data and methodology

The authors clearly did a great deal of work to identify VMAT2 proteins amenable to cryo-EM structure determination including screening expression and stability of the protein from twelve species. They attempted to solve the cryo-EM structure of the wild-type protein, but initially failed due to the small size and featureless structure of the detergent solubilized protein. The authors then generated VMAT2 constructs with BRIL or AmpC inserted into various loops to increase the size and enhance the shape of the molecule to aid in particle alignment. One of the sheep VMAT2-BRIL constructs, OaVMAT2TM8/9-BRIL, was used for single particle cryo-EM data collection and structure determination that resulted in a 3.2-Angstrom map. However, the map revealed a dimeric structure with an unexpected monomer fold and an unexpected dimer interface based on comparison with a previous structure of a homo-dimeric MFS transporter (hMCT2, ref 11). The authors went on to carefully analyze the substrate and inhibitor binding properties of OaVMAT2TM8/9-BRIL and performed computational docking to determine whether the unexpected fold of the cryo-EM structure was consistent with the properties of the WT protein or an artifact of the BRIL insertion. The authors concluded that the engineered protein was misfolded.

The authors then shifted their focus back to cryo-EM analysis of WT type VMAT2 proteins (human, sheep, and frog) and, remarkably, obtained a 4-Angstrom reconstruction of frog VMAT2, XIVMAT2,

that revealed the structure of the TM domain and some loops. This apo structure was found to be in the lumen-facing conformation and the authors describe a cytosolic gate comprised of three methionine and one tyrosine residue. The authors computationally docked substrate (serotonin) and inhibitor (reserpine) to the cryo-EM structure and describe the details of the binding sites. They use the substrate and inhibitor docking results to mutate and biochemically validate the binding sites in human VMAT2, and their mutagenesis and binding studies supported the identities of the binding site residues.

The authors compared their apo XIVMAT2 structure with the recently solved substrate- and inhibitor-bound structures of human VMAT2 and found that their structure was similar to serotonin-bound VMAT2 (Refs 17 and 18). They also found that their computational docking of serotonin was supported by the serotonin-bound cryo-EM structures. However, they found that the docking results with the inhibitor reserpine were significantly different from a cryo-EM structure of reserpine-bound human VMAT2 (with MBP and MBP-specific DARPin fused to the N- and C-termini respectively) (Pidathala et al. Ref 16).

The quality and thoroughness of the work, analysis, and presentation are excellent and may be of general interest to technical structural biologists. However, a substantial portion of the manuscript describes analysis of the mis-folded OaVMAT2TM8/9-BRIL which is only valuable as a cautionary tale (as intimated by the manuscript title) and the remainder focuses on the apo XIVMAT2 structure that has been superseded by higher-resolution structures of human VMAT2 in a variety of mechanistically revealing structural states.

Analytical approach

The analyses were thorough and considered available relevant data.

Suggested improvements

The authors could try to solve the cryo-EM structures of wild-type serotonin- and reserpine-bound XIVMAT2 (or preferably human VMAT2) to determine whether the protein engineering used in Refs 16 and 17 altered the structures.

Clarity and context

The manuscript was well written and presented and includes analysis of work published during manuscript preparation.

References

Appropriate references were included and discussed.

Reviewer #2 (Remarks to the Author):

The work by Lyu et al. presents two cryo-EM structures of VMAT2, featuring BRIL-incorporated sheep VMAT2 in an atypical dimer form, and frog VMAT2 in a lumen-facing conformation adopting a canonical MFS fold. The authors also use molecular docking and MST binding experiments to provide some insights into the mechanisms of serotonin and reserpine, although these insights do not significantly extend beyond what has been previously shown in published structural/functional works on human VMAT2. This work may still be useful to the field of VMAT2. However, some more experiments are needed to strengthen the paper. I have the following suggestions:

1. It is important to demonstrate the transport activity of the constructs compared to the wild type, particularly the OaVMAT2TM8/9-BRIL variant, as substrate binding does not always correlate with transport activity. If the construct is not functional, it is hard to see value in analyzing the atypical MFS structure. It is well known that construct modification can lead to adverse effects, including changes in protein fold. For example, a single amino acid mutation in TRPV5 (PMID: 28878326) can change its fold from swapping to non-swapping. For VMAT2, it is true that other groups do use engineered constructs, but they all demonstrated that the constructs used have transport activity.

2. The authors stated that they also studied wild-type OaVMAT2 (line 112) but did not disclose its oligomeric state. It is important for the authors to clarify whether the wild-type is a monomer or a dimer. If the wild-type is a monomer, it should be clearly stated that the dimerization is a pure artifact, instead of using vague language like “hypothesize” or “may” in the sentence, “We hypothesize that replacement of the TM8/9 loop with BRIL may have altered OaVMAT2 folding.”
3. The fused BRIL looked far away from the dimerization interface. How does it contribute to the dimer formation? Or does the author know if it is the fused BRIL or the deletion of the linker that induces dimerization?
4. In line 58, the authors mentioned that the AmpC-incorporated construct (OaVMAT2TM8/9-AmpC) is also stable for cryo-EM analysis. Does this construct also form a dimer?
5. What is the sequence conservativity of the dimerization interface? If the residues mediating dimerization are conserved, why was the dimerized form not observed in frog VMAT2 or in reported cases (human VMAT2)? This should be mentioned and discussed.
6. What is the rationale for using different SEC buffers for sheep (Tris pH 7.5) and frog (MES pH 6.0) VMAT2? The buffer pH change may induce different conformations for proton-driven transporters. Are the different conformations observed in sheep and frog VMAT2 caused by the pH? This should be addressed.
7. Why was the MES pH 6.0 buffer used for the MST measurement? Is the binding consistent between MES 6.0 and Tris 7.5 buffers?
8. Reserpine inhibits VMAT2 in its cytosol-facing conformation. Could the authors elaborate on the reason why they used a lumen-facing model (frog VMAT2) to do docking analysis for reserpine?

REVIEWER COMMENTS

Reviewer #1 (Remarks to the Author):

Key results

The authors solved the single particle cryo-EM structure of sheep VMAT2 with BRIL inserted into a loop to increase the molecular size and enhance the shape for structure determination. They obtained a high-quality cryo-EM reconstruction of the chimeric protein but found that the VMAT2 fold was significantly different than expected based on previous major facilitator superfamily (MFS) structures. The authors found that serotonin and dopamine bound the engineered protein with affinities similar to that of the wild-type protein. However, the VMAT2 inhibitor reserpine bound to the wild-type protein with high affinity but did not bind to the engineered protein, which confirmed that the unexpected fold was an artifact of the protein engineering. The authors succeeded in obtaining a 4-Angstrom cryo-EM map of apo, wild-type frog VMAT2 which revealed the expected MFS fold. The authors performed site-directed mutagenesis and associated serotonin and reserpine binding studies on both constructs and conducted computational docking with these molecules and their cryo-EM structures. During preparation of this manuscript, several structural studies on VMAT2 were published including Wu et al., Nature 2023 that reported four structures of an engineered human VMAT2 each bound either substrate or an inhibitor (Ref 17) and Pidathala et al. that reported structures an engineered human VMAT2 with inhibitor bound (Ref 16). The authors compared their wild-type, apo frog VMAT2 structure and computational docking results with the recently published VMAT2 structures, and they made the case that their work demonstrates that protein engineering to aid in structure determination may result in well-folded but physiologically irrelevant structures.

Response:

Thanks for your comment. In the revised manuscript, we have also employed a cell-based uptake assay to demonstrate transport ability of OaVMAT2_{TM8/9-BRIL}, which retains ~62% of transport activity compared to wild-type OaVMAT2 (Fig. 1g and Table S3). This data further suggests that extra caution may be necessary when interpreting structures of engineered proteins.

Validity

The authors performed careful cryo-EM structure determination and thorough structure-based mutagenesis and biochemical analysis. They used computational docking rather than

experimentally determined structures to gain insight into substrate and inhibitor binding. They described similarities and differences between their docking results and related experimentally determined co-structures.

Response:

Thanks for your comment.

Significance

It is a good point that protein engineering for structure determination should be used with caution, and the successful generation of the ~4-Angstrom map of wild-type frog VMAT2 by single particle cryo-EM is technically impressive. However, the publications describing experimentally-determined apo, substrate-, and inhibitor-bound human VMAT2 structures that were noted and analyzed by the authors (Refs 15 – 18) greatly diminished the significance of the structural work presented here.

Response:

Thanks for your comment. Indeed, it's unfortunate that our first VMAT2 structure (OaVMAT2_{TM8/9-BRIL}) was in an atypical fold, which took us more time to address related issues. Certainly, it's a bit disappointing to lose a race. But we understand that's how science advances, and we are making our best effort to contribute to this process.

Data and methodology

The authors clearly did a great deal of work to identify VMAT2 proteins amenable to cryo-EM structure determination including screening expression and stability of the protein from twelve species. They attempted to solve the cryo-EM structure of the wild-type protein, but initially failed due to the small size and featureless structure of the detergent solubilized protein. The authors then generated VMAT2 constructs with BRIL or AmpC inserted into various loops to increase the size and enhance the shape of the molecule to aid in particle alignment. One of the sheep VMAT2-BRIL constructs, OaVMAT2_{TM8/9-BRIL}, was used for single particle cryo-EM data collection and structure determination that resulted in a 3.2-Angstrom map. However, the map revealed a dimeric structure with an unexpected monomer fold and an unexpected dimer interface based on comparison with a previous structure of a homo-dimeric MFS transporter (hMCT2, ref 11). The authors went on to carefully analyze the substrate and inhibitor binding properties of OaVMAT2_{TM8/9-BRIL} and performed computational docking to determine whether the

unexpected fold of the cryo-EM structure was consistent with the properties of the WT protein or an artifact of the BRIL insertion. The authors concluded that the engineered protein was misfolded.

The authors then shifted their focus back to cryo-EM analysis of WT type VMAT2 proteins (human, sheep, and frog) and, remarkably, obtained a 4-Angstrom reconstruction of frog VMAT2, XIVMAT2, that revealed the structure of the TM domain and some loops. This apo structure was found to be in the lumen-facing conformation and the authors describe a cytosolic gate comprised of three methionine and one tyrosine residue. The authors computationally docked substrate (serotonin) and inhibitor (reserpine) to the cryo-EM structure and describe the details of the binding sites. They use the substrate and inhibitor docking results to mutate and biochemically validate the binding sites in human VMAT2, and their mutagenesis and binding studies supported the identities of the binding site residues.

The authors compared their apo XIVMAT2 structure with the recently solved substrate- and inhibitor-bound structures of human VMAT2 and found that their structure was similar to serotonin-bound VMAT2 (Refs 17 and 18). They also found that their computational docking of serotonin was supported by the serotonin-bound cryo-EM structures. However, they found that the docking results with the inhibitor reserpine were significantly different from a cryo-EM structure of reserpine-bound human VMAT2 (with MBP and MBP-specific DARPIn fused to the N- and C-termini respectively) (Pidathala et al. Ref 16).

The quality and thoroughness of the work, analysis, and presentation are excellent and may be of general interest to technical structural biologists. However, a substantial portion of the manuscript describes analysis of the mis-folded OaVMAT2_{TM8/9-BRIL} which is only valuable as a cautionary tale (as intimated by the manuscript title) and the remainder focuses on the apo XIVMAT2 structure that has been superseded by higher-resolution structures of human VMAT2 in a variety of mechanistically revealing structural states.

Response:

Thanks for your in-depth comment. In the revised manuscript, we have used a cell-based uptake assay to demonstrate that OaVMAT2_{TM8/9-BRIL} retains ~62% of transport activity compared to wild-type OaVMAT2 (Fig. 1g and Table S3). This data suggests that OaVMAT2_{TM8/9-BRIL} is intriguing not only for its atypical structural fold as an MFS transporter, but also for its being functional with such a fold. Certainly, evaluation of the biological relevance of this atypical fold of OaVMAT2_{TM8/9-BRIL} requires further investigation. This description has been added in the Results section.

Analytical approach

The analyses were thorough and considered available relevant data.

Response:

Thanks for your comment.

Suggested improvements

The authors could try to solve the cryo-EM structures of wild-type serotonin- and reserpine-bound XIVMAT2 (or preferably human VMAT2) to determine whether the protein engineering used in Refs 16 and 17 altered the structures.

Response:

Thanks for your comment. As suggested, we have prepared human VMAT2 (hVMAT2) samples in the presence of 10 mM dopamine and collected 7,650 cryo-EM movie stacks using a Titan Krios. Unfortunately, after many rounds of data processing, we were only able to obtain a volume of ~6.5 Å (Fig. R1), which is probably limited by raw data quality, raw image quantity and time. Due to the resolution limit, we were not able to analyze the hVMAT2-dopamine structure to extract useful structural information. This task will require further investigation.

Figure R1. Workflow of cryo-EM data processing of hVMAT2_{WT}-dopamine dataset.

Clarity and context

The manuscript was well written and presented and includes analysis of work published during manuscript preparation.

Response:

Thanks for your comment.

References

Appropriate references were included and discussed.

Response:

Thanks for your comment.

Reviewer #2 (Remarks to the Author):

The work by Lyu et al. presents two cryo-EM structures of VMAT2, featuring BRIL-incorporated sheep VMAT2 in an atypical dimer form, and frog VMAT2 in a lumen-facing conformation adopting a canonical MFS fold. The authors also use molecular docking and MST binding experiments to provide some insights into the mechanisms of serotonin and reserpine, although these insights do not significantly extend beyond what has been previously shown in published structural/functional works on human VMAT2. This work may still be useful to the field of VMAT2. However, some more experiments are needed to strengthen the paper. I have the following suggestions:

1. It is important to demonstrate the transport activity of the constructs compared to the wild type, particularly the OaVMAT2_{TM8/9-BRIL} variant, as substrate binding does not always correlate with transport activity. If the construct is not functional, it is hard to see value in analyzing the atypical MFS structure. It is well known that construct modification can lead to adverse effects, including changes in protein fold. For example, a single amino acid mutation in TRPV5 (PMID: 28878326) can change its fold from swapping to non-swapping. For VMAT2, it is true that other groups do use engineered constructs, but they all demonstrated that the constructs used have transport activity.

Response:

Thanks for your insightful comment and suggestion. In the revised manuscript, we have employed a cell-based uptake assay to measure transport activities of various VMAT2 constructs (Fig. 1g and Table S3). Interestingly, OaVMAT2_{TM8/9-BRIL} showed ~62% of transport activity compared to wild-type OaVMAT2 (Fig. 1g and Table S3), suggesting that OaVMAT2_{TM8/9-BRIL} is a (partially) functional MFS transporter with an atypical fold. Certainly, evaluation of the biological relevance of this atypical fold of OaVMAT2_{TM8/9-BRIL} requires further investigation. Meanwhile, this data further suggests that extra caution may be necessary when interpreting structures of engineered proteins. This description has been added in the Results and Discussion sections.

2. The authors stated that they also studied wild-type OaVMAT2 (line 112) but did not disclose its oligomeric state. It is important for the authors to clarify whether the wild-type is a monomer or a dimer. If the wild-type is a monomer, it should be clearly stated that the dimerization is a

pure artifact, instead of using vague language like “hypothesize” or “may” in the sentence, “We hypothesize that replacement of the TM8/9 loop with BRIL may have altered OaVMAT2 folding.”

Response:

Thanks for your comment and question. The wild-type OaVMAT2 appears a monomer in detergent solutions as revealed by a chemical crosslinking experiment (no dimer band) (Fig. S1e). In the original manuscript, the sentence “We hypothesize that replacement of the TM8/9 loop with BRIL may have altered OaVMAT2 folding” was meant to describe different positions of TM11 and TM12 (compared to a canonical MFS fold, Fig. S4d) that may be caused by the insertion of BRIL in the TM8/9 loop. And as a result, TM11 and TM12 (together with TM5 and TM8) participated directly in formation of the dimerization interface between two OaVMAT2_{TM8/9-BRIL} monomers (Fig. 1a). As suggested, we have added description about the oligomeric state of wild-type OaVMAT2 in the revised manuscript (Discussion section) and edited the text to clearly state that the oligomeric state of wild-type OaVMAT2 is different from that of OaVMAT2_{TM8/9-BRIL}, which is likely an artifact induced by its atypical fold and engineering (Discussion section).

3. The fused BRIL looked far away from the dimerization interface. How does it contribute to the dimer formation? Or does the author know if it is the fused BRIL or the deletion of the linker that induces dimerization?

Response:

Thanks for your comment and question. Indeed, the fused BRIL is located quite far away from the dimer interface. Therefore, it looks less likely that the fused BRIL (or the deletion of the TM8/9 loop) caused dimerization directly. Rather, in our opinion, insertion of BRIL in the TM8/9 loop may have caused different placement of TM11 and TM12 in OaVMAT2_{TM8/9-BRIL} compared to a canonical MFS fold (Fig. S4d). As a result, TM11 and TM12 (together with TM5 and TM8) participated directly in formation of the dimerization interface between two OaVMAT2_{TM8/9-BRIL} monomers (Fig. 1a). Therefore, the fused BRIL may have contributed indirectly to the dimer formation through TM11 and TM12.

4. In line 58, the authors mentioned that the AmpC-incorporated construct (OaVMAT2_{TM8/9-AmpC}) is also stable for cryo-EM analysis. Does this construct also form a dimer?

Response:

Thanks for your question. Based on chemical crosslinking experiments, a minor dimer band of OaVMAT2_{TM8/9-AmpC} in detergent solutions was observed in SDS-PAGE analysis, whereas the majority of OaVMAT2_{TM8/9-AmpC} shows in the monomer band (or in the high-molecular-weight aggregate band) (Fig. S1e). The chemical crosslinking data has been added in the Discussion section.

5. What is the sequence conservativity of the dimerization interface? If the residues mediating dimerization are conserved, why was the dimerized form not observed in frog VMAT2 or in reported cases (human VMAT2)? This should be mentioned and discussed.

Response:

Thanks for your question and suggestion. The residues that form the dimer interface of OaVMAT2_{TM8/9-BRIL} are highly conserved among different species (residues highlighted in yellow in Fig. S8). However, in a canonical MFS fold, many of these residues would not be exposed on the protein surface due to different positions of TM11 and TM12. For example, in XIVMAT2_{WT}, residues Y417, Y421, A424 and F428 on TM11 (equivalent to Y421, Y425, A428 and F432 in OaVMAT2) are facing the interior of VMAT2 and are not available for dimerization (Fig. S9b). This result may explain why OaVMAT2_{TM8/9-BRIL} forms a dimer but XIVMAT2_{WT} does not. This description has been added in the Discussion section.

6. What is the rationale for using different SEC buffers for sheep (Tris pH 7.5) and frog (MES pH 6.0) VMAT2? The buffer pH change may induce different conformations for proton-driven transporters. Are the different conformations observed in sheep and frog VMAT2 caused by the pH? This should be addressed.

Response:

Thanks for your question and suggestion. The VMAT2 samples were prepared in different pH buffers intentionally, trying to capture different conformations of VMAT2. For example, frog VMAT2 (XIVMAT2) was actually prepared in both pH 6.0 (mimicking the vesicular lumen pH) and pH 7.5 (mimicking the cytosol pH) buffers for cryo-EM data collection, and we were hoping to solve two different conformations with these two pH conditions. But unfortunately, we were only able to solve the XIVMAT2 structure at ~4 Å with the pH 6.0 data set.

Meanwhile, the sheep VMAT2 structure (OaVMAT2_{TM8/9-BRIL}, pH 7.5) and the frog VMAT2 structure (XIVMAT2_{WT}, pH 6.0) adopt different conformations (different folds) (Fig. S7b).

However, their difference is more likely caused by the fused BRIL in OaVMAT2_{TM8/9-BRIL} rather than different pHs, thus not suitable for discussing the pH effect on VMAT2 conformations. Therefore, in the revised manuscript, we have tested resistance of purified XIVMAT2 protein to digestive enzymes (e.g. trypsin or chymotrypsin) at different pHs. The XIVMAT2 protein sample showed distinct enzymatic digestion patterns at pH 6.0 vs. pH 7.5 (Fig. S9c), suggesting that XIVMAT2 likely assumes different conformations at these two pHs. This result and description has been added in the Discussion section.

7. Why was the MES pH 6.0 buffer used for the MST measurement? Is the binding consistent between MES 6.0 and Tris 7.5 buffers?

Response:

Thanks for your question. MST binding results are largely consistent between MES pH 6.0 and Tris pH 7.5 buffers (e.g. DOP/SER for HsVMAT2), but with a stronger binding signal at pH 6.0 than at pH 7.5 (Fig. R2). This is the reason that the MST measurements were performed in the MES pH 6.0 buffer in this work.

Figure R2. MST fitting curves of DOP (panel a) and SER (panel b) binding to wild-type HsVMAT2 at pH 6.0 (black dots) and 7.5 (red triangles), $N=3$.

8. Reserpine inhibits VMAT2 in its cytosol-facing conformation. Could the authors elaborate on the reason why they used a lumen-facing model (frog VMAT2) to do docking analysis for reserpine?

Response:

Thanks for your comment and question. We were aware that reserpine binds more favorably in the cytosol-facing conformation of VMAT2. However, with the two VMAT2 structures (OaVMAT2_{TM8/9-BRIL} and XIVMAT2_{WT}) in our hands, only the lumen-facing XIVMAT2_{WT} structure has a central cavity that is large enough for reserpine docking analysis. Interestingly, a molecular dynamics simulation study has shown that reserpine displays high favorable binding enthalpies in both cytosol-facing and lumen-facing models of VMAT2 (Støve et al., *Commun Biol*, 2022). Therefore, we performed a docking analysis for reserpine using the lumen-facing XIVMAT2_{WT} structure in the original manuscript.

Meanwhile, we realized that the docking analysis of reserpine in the lumen-facing XIVMAT2_{WT} structure added only limited information and value to the main findings and conclusions of this work, while it may introduce potential confusions as pointed out by your comment. Therefore, the reserpine docking analysis has been removed from the revised manuscript.

REVIEWERS' COMMENTS

Reviewer #1 (Remarks to the Author):

I have no further questions or concerns.

Reviewer #2 (Remarks to the Author):

The authors have satisfactorily addressed my questions. I have no further requests.

REVIEWERS' COMMENTS

Reviewer #1 (Remarks to the Author):

I have no further questions or concerns.

Response:

Thanks for your comment.

Reviewer #2 (Remarks to the Author):

The authors have satisfactorily addressed my questions. I have no further requests.

Response:

Thanks for your comment.